

# Online painting image clustering for the mental health of college art students based on improved CNN and SMOTE

Fake Ma and Huwei Li

Henan Economy and Trade Vocational College, Zhengzhou, China

## ABSTRACT

In modern education, mental health problems have become the focus and difficulty of students' education. Painting therapy has been integrated into the school's art education as an effective mental health intervention. Deep learning can automatically learn the image features and abstract the low-level image features into high-level features. However, traditional image classification models are prone to lose background information, resulting in poor adaptability of the classification model. Therefore, this article extracts the lost colour of painting images based on K-means clustering and proposes a painting style classification model based on an improved convolutional neural network (CNN), where a modified Synthetic Minority Oversampling Technique (SMOTE) is proposed to amplify the data. Then, the CNN network structure is optimized by adjusting the network's vertical depth and horizontal width. Finally, a new activation function, PPReLU, is proposed to suppress the excessive value of the positive part. The experimental results show that the proposed model has the highest accuracy in classifying painting image styles by comparing it with state-of-the-art methods, whose accuracy is up to 91.55%, which is 8.7% higher than that of traditional CNN.

## INTRODUCTION

Art is the earliest communication language of human beings, symbolizing human thoughts, emotions, reality, and imagination. As one of the most common forms of artistic expression, painting has been widely used in many disciplines. The university stage is crucial for a person to mature from immaturity and drastic psychological changes gradually. More attention should be paid to his mental health (*Sun & Liu, 2022*). Especially since 2020, under the impact of the COVID-19 epidemic, some college students have shown prominent characteristics of unstable psychological states and quickly fluctuating emotions, and their overall mental health level is not good. Modern society is in a period of rapid development. The application of painting art therapy to college students' mental health education can provide new ideas for improving college students' mental health literacy but also help to meet the diverse psychological service needs of college students and help improve the level of college students' mental health.

Corresponding author
Huwei Li, Lhw861108@126.com

Paintings, however, are difficult to obtain, making researchers' research work more difficult. In the mobile Internet era, people's lives are being changed by digital technology, paintings are digitized in the form of images, and many electronic databases of paintings are being gradually established. The digitization of paintings enables researchers to obtain a large amount of painting data through the Internet, promoting research progress in this field (*Adachi, Natori & Aikawa, 2021*). Natural image is mainly an accurate description of the real scene, and its content is slightly different from the actual scene, while painting images are highly abstract and general. It is a conceptual generalization of real things by painters, and its characteristics are highlighted and exaggerated. Because of subjective characteristics such as the painter's style and artistic style, its content differs from the actual scene (*Ning et al., 2022*; *Zhong, Huang & Xiao, 2020*). Traditional image classification methods must rely on experienced researchers to manually extract complex image features. This feature extraction method not only consumes a lot of workforces but also quickly loses many detailed features, which leads to the poor adaptability of classification models to test data sets (*Sun et al., 2015*).

In recent years, the CNN method has made exemplary achievements in computer vision (*Cornelis et al., 2016*). It can automatically extract and learn image features, and its model generalization ability is strong. For example, *Sun et al. (2015)* proposed a CNN method based on mixed sparseness to extract the brushwork features of Chinese ink painting, classified them according to authors, and achieved good results. *Sun et al. (2016)* proposed a method based on multi-scale directional decomposition and controllable pyramid to extract different paintings' multi-scale and spatial variation features. The feature analysis found that artists have their own painting style features so that computer technology can be used to classify the works of different writers and transfer the styles of paintings. CNN can improve the accuracy of classification or prediction. Still, the number of data sets of painting images is limited and insufficient to support the model to complete practical training, so other schemes are needed to expand the data sets (*Jiang et al., 2006*).

Traditional image classification methods must rely on experienced researchers to manually extract complex image features. Such feature extraction methods not only cost a lot of human resources but also quickly lose many detailed features. Thus, the classification model is not adaptable to test data sets. At the same time, research on colour feature extraction and classification in complex paintings and their application in mental health education are scarce. Therefore, the main contributions of this article are as follows:

First, to solve the overfitting problem, an improved SMOTE amplification data was proposed, in which several values were inserted into a few relatively close samples. Then, the network's longitudinal depth and transverse width are adjusted to optimize the structure of the CNN network. Finally, a new activation function PPReLU is proposed, which introduces learnable parameters in the positive part to suppress the excessive value of the positive part. The method in this article realizes the extraction of multiple-size features, which can more effectively extract the elements of painting images and complete image classification, learn the effective extraction of different painting styles, and contribute to the realization of painting art therapy.

This article first reviews the application of painting image classification and CNN in the context of deep learning. Then, it extracts the lost colour of painting images based on K-means clustering and proposes a painting style classification model based on improved CNN. Finally, we amplify and train the Gallery of Aesthetic Landscape Oil Paintings dataset and compare the training results of different activation functions and deep learning algorithms.

# RELATED WORKS

## Classification of painting images

Chinese painting is a kind of image understanding in unnatural scenes, and its semantic information is richer and more abstract than traditional digital images. However, its source is limited and there are some problems, such as its distribution is not concentrated, its classification is not clear, its collection is not easy, and the number of each category is not large (*Liu & Jiang, 2014*). Traditional painting image classification research is mostly based on shallow learning. *Hua et al. (2020)* describes the granularity and sparsity of image edges by extracting the edge size histogram and classifying them by support vector machines (SVM). By extracting colour and texture features, *Kang, Shim & Yoon (2018)* adopted Bayes, Fisher's linear discriminant analysis (FLD) and SVM to make a variety of classifiers, such as painting techniques and authors. These learning algorithms are based on shallow structure. Although some achievements have been made, limitations exist, such as limited computing power and samples. When dealing with complex classification problems, the generalization ability is limited, which requires manual design and feature extraction (*Li, Guo & Ren, 2018*). Some studies try to extract the low-level features of images and analyze image emotions by combining the professional knowledge of human psychology or visual observation cognition. *Ciregan, Meier & Schmidhuber (2012)* constructed a data set of colour-matching emotion words based on the relationship between colour and emotion. Then they predicted the feeling of paintings by creating a colour spectrum for Western paintings and finding the corresponding emotion words in the data set. *Zhang, Zhao & LeCun (2015)* used a weighted K nearest neighbour distance algorithm to predict subjective and fuzzy emotions in abstract paintings by extracting image colour and texture features.

## Application of CNN

Convolutional neural network (CNN) avoids complicated image processing in the early stage. By automatically learning features from a large amount of data, it can more accurately represent the rich information hidden in the data and has made outstanding achievements in many fields. *Tuggener, Schmidhuber & Stadelmann (2021)* adopted the CNN method to identify numbers in the MNIST database, which achieved the best results compared with other methods, with an error rate of less than 0.3%. *Zhou et al. (2019)* compared CNN with multiple algorithms in document character recognition and verified that CNN is superior to all other algorithms. *Zheng & Zhang (2018)* applied deeper CNN to solve the problem of ImageNet, which significantly promoted the development of image recognition. Many researchers introduce CNN features into the field of automatic analysis of Chinese computer paintings to acquire deep convolution features and analyze more semantic features. *Sheng*

*& Li (2018)* combined the bottleneck layer idea of deep convolutional encoder–decoder and Inception module in GoogLeNet, reduced the model parameters and accelerated the calculation speed, and obtained the same performance of ink painting style transfer rendering as that of the network before improvement. *Li, Sheng & Hua (2018)* proposed a method to simulate the creation process of colour ink painting, using CNN and Generative Adversarial Networks (GAN) to convert lines and colour styles and stylize flower photos into colour ink paintings. *Shen et al. (2018)* used CNN to extract the high-level semantic features of artistic goals depicted in Chinese paintings. They introduced a support vector machine to fuse the classification results of each goal to realize the recognition of Chinese painting authors. *Lasheng & Yuqiang (2017)* put forward an improved embedded learning algorithm, which extracts convolution features of traditional Chinese painting using the fine-tuned VGG model, and identified the author of traditional Chinese painting by the influence of feature selection and the importance of traditional Chinese painting features on classification.

The above methods aim at the creation of the paintings. The complicated painting colour feature extraction and classification of the research are very rare at the same time. CNN has improved the classification or prediction accuracy because it can be trained through vast amounts of data to build a model containing many hidden layers. However, the number of data sets of painting images is limited and insufficient to support the model to complete the effective training. Therefore, other schemes are needed to expand the data sets.

# COLOUR EXTRACTION BASED ON K-MEANS CLUSTERING

In the field of image colour extraction, to use quantitative data information to reflect colour, some scholars have introduced computer image processing technology based on clustering algorithms in their research (*Wu, Zhang & Yuan, 2019*). In this section, the lost colour information in the repaired image is extracted, and then K-means are optimized by the elbow method to carry out colour clustering.

## Colour loss information extraction

Select the colour distribution in the complete painting area and the colour overlapping and matching texture. The Criminisi algorithm extracts the lost colour information in the restored image. At the same time, according to the colour texture in the original image, the superposition instruction is executed to obtain the required fixed colour. The first step of the algorithm is to extract the positioning node, set the priority of each node in the whole colour part, find out the node position with the highest weight proportion, and take this node as the colour extraction centre to extract the local colour of painting in a small area. The calculation formula of the node priority is:

$$P\left(p_i\right) = \frac{C\left(p_i\right)D\left(p_i\right)}{\gamma\mu^{-i}} \tag{1}$$

Where: $p_i$ represents the specific position of the most characteristic colour data at i; $C\left(p_i\right)$ represents the confidence of the data; $D\left(p_i\right)$ indicates the degree of support for this data;

**Peer**J Computer Science

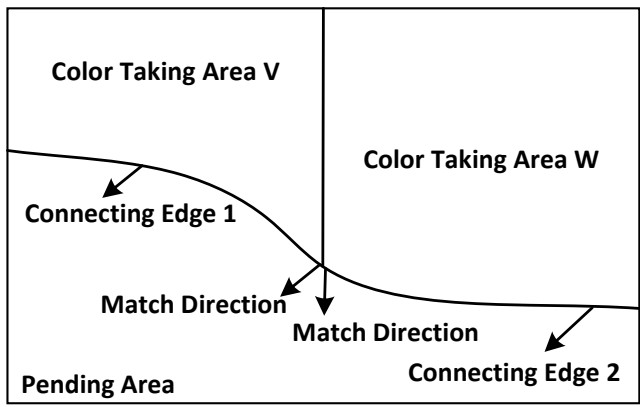

**Figure 1** Extraction area of color loss.

$\gamma$ represents the iteration number of the system; $\mu^{-i}$ represents the position correction parameter. The schematic diagram of regional colour extraction is shown in Fig. 1.

According to different positions, colours with the same colour are extracted, and colours are superimposed according to the colour texture trend to synthesize the colours with the same original colour, thus realizing the colour extraction and synthesis of painting restoration.

## Colour clustering

K-means is widely used in clustering algorithms because of its simple principle and strong operability. However, selecting the initial cluster centre is challenging, and the K value of the number of clusters is difficult to determine, so it needs to be optimized first.

When traditional clustering algorithms cluster images in colour, there are two ways to choose the initial clustering centre: hue mode and grey mode. Hue mode means that the initial cluster centre is selected by hue and distributed evenly along hue H * from 0 to 360 according to the desired value of K. Gray mode means that the initial cluster centre is selected by grey value, and the grey value is evenly distributed according to the required K value from 0 to 255. Both forms have a defect; they cannot reasonably change the initial cluster centre according to the difference in image pixel composition. Therefore, this article uses the max–min criterion method to optimize the selection of the initial cluster centre. First in the data set, randomly selected from a point as the initial clustering centre $v_1$, calculate the distance from the first cluster centre to the furthest point as the second $v_2$ clustering centre, from the rest of the points calculation to the first two clustering centres of the Euclidean distance minor point and put in a set V, the most extensive collection of perimeter points as the third clustering centre. The formula repeats the calculation until the maximum and minimum distance is not greater than the distance of the first and second cluster centres.

$$\text{dist}_l = \max\{\min(\text{dist}_{i1}, \text{dist}_{i2}, \cdots)\}$$
$$(l, i = 1, 2, \cdots, n) \tag{2}$$

where $dist_{i1}$ and $dist_{i2}$ respectively represents the Euclidean distance from sample i to $v_1$ and $v_2$, respectively.

In different times and geographical environments, because the colour-matching styles of paintings will be different, the optimized way of selecting the initial clustering centre can better select different initial clustering centres according to the pixel distribution of different painting images and improve the accuracy of colour extraction.

## Number of class clusters

Traditional clustering algorithms generally need to rely on personal experience to input the estimated value of the number of clusters K, then adjust the K value by comparing the image segmentation effect under different cluster numbers. It is only suitable for experiments with small sample sizes. To obtain the best number of clusters of experimental objects, reduce manual input and improve experimental efficiency, the elbow method is introduced to estimate the number of clusters. The elbow method calculates the sum of squared error (SSE) of the sample points of each class and takes it as a measure. The smaller the value, the more convergent each class cluster is

$$SSE = \sum_{i=1}^{k} \sum_{p \in L_i} \|p - q_i\|^2 \tag{3}$$

Where: p represents the data object in the i-th class group $L_i$, $q_i$ represents the mean value of all data objects in a class group, and k represents the number of classification groups.

Select a sample image from the sample set as the measured image for the experiment. Firstly, a pixel in the background colour area of the image is identified. The set of all pixels with the same colour characteristics as the pixel in the image is regarded as the image's background colour. The influence of the image's background colour is eliminated, and the elbow method estimates the K value. As shown in Fig. 2, when k is 5, the curve has an obvious falling inflexion point, which is consistent with the expected results of the experiment. It is verified that the elbow method can help to determine the best number of clusters k value of image clustering.

## CLASSIFICATION OF PAINTING STYLES BASED ON CNN

The reason why CNN can improve the accuracy of classification or prediction is that it can construct a model with multiple hidden layers by training massive data to learn the features that can better represent the data. Due to the limitation of the number of painting images, an improved SMOTE method is adopted in this article to expand the data set.

## Improved SMOTE

SMOTE is the main principle of inserting some values into a few samples at a close distance to generate a small number of new class samples to increase the number of small samples and improve the classification accuracy of small class samples. However, the samples generated by this method will be scattered in dense areas, which cannot solve the problem of sparse distribution of current experimental models. The improved SMOTE algorithm proposed in this article properly fills the sparse areas between classes. Firstly, each class of

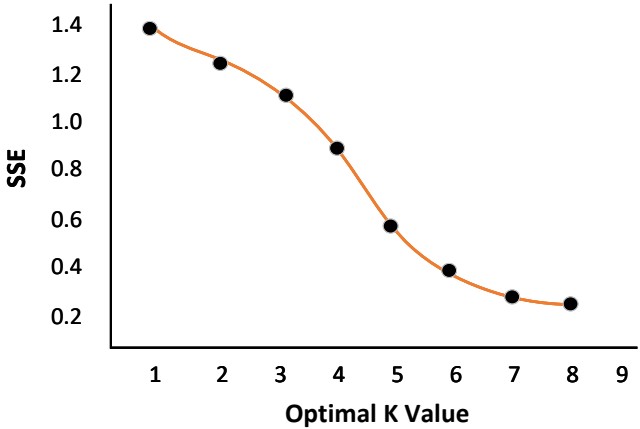

**Figure 2** **Determination of the number of class clustersk.** (A) Measured Image. (B) Optimal K value.

samples is clustered separately, and each class of samples is grouped into k classes. Then, random interpolation is performed on each existing sample on the path to a random sample in each cluster so that the original samples of each class are expanded by k times, and the generated sample set is k+1 times the original sample set. Adjust the value of k according to the size of the required sample set. For example, if the original sample set is n, double the original sample set and set k to 1. This method can better prevent over-fitting than adding samples such as noise and random copying. Although adding noise, random copying, and other methods increase the number of samples to a certain extent, this method also adds some additional training data, which makes the construction time of the classifier longer. On the other hand, the added class samples do not contain much useful information, so it cannot solve the over-fitting problem well.

## CNN network structure

The selection of CNN's network structure will affect the performance of the classification results. Many factors affect its classification, such as the size of the input data, the network's depth selection and the training parameters' setting. This article mainly optimizes the network by adjusting the network structure of CNN, that is, adjusting the network's vertical depth and horizontal width. On the depth, adjust the stacking number of the convolution layer and sub-sampling layer, while in terms of width, the number of characteristic graphs of each layer of the network and the size of each layer plane are adjusted. After many adjustments, the input image size of CNN is finally selected to be $32 \times 32$. The best structure is six layers: one layer of input, two layers of convolution, two layers of sub-sampling and one layer of full connection as shown in Fig. 3.

## Activation function

The activation function is an indispensable part of the neural network, which is used to introduce nonlinear factors to enhance the expression ability of the neural network model. The activation function of CNN is generally a ReLU function, and its mathematical expression is Max (0,x). The sparse matrix is established to simplify the forward feature

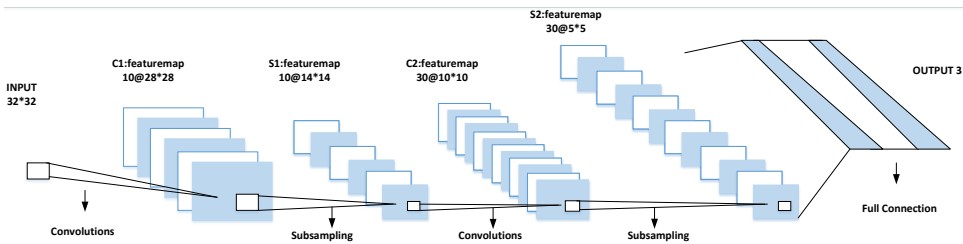

**Figure 3** Classification accuracy under different activation functions.

extraction and reverse weight update calculation, thus speeding up the calculation speed of the network. The expression is shown in Eq. (4).

$$f(y_i) = \begin{cases} 0, y_i \leq 0 \\ y_i, y_i > 0 \end{cases} \quad (4)$$

The positive gradient of ReLU is 1, which somewhat solves the problem of gradient disappearance of deep neural networks. However, the negative gradient is 0, which still has the problem of gradient disappearance. The activation function PReLU is an improved version of ReLU that the positive part gradient is 1, and the negative part gradient is a learnable parameter, which effectively solves the problem that the negative part gradient disappears in ReLU. However, the positive part still has the problem of excessive value. The formula of the PRELU activation function is defined as:

$$f(y_i) = \begin{cases} a_i y_i, y_i \leq 0 \\ y_i, y_i > 0 \end{cases} \quad (5)$$

Given the problems in the PReLU activation function, a new activation function PPReLU is proposed. A learnable parameter is introduced into the positive part to suppress the excessive value of the positive part. According to the numerical distribution of the positive part of the PReLU activation function, the numerical value is mainly distributed between 0 and 4, so the piecewise threshold of the positive part of the activation function PPReLU is selected as 4. The activation function formula of PPReLU is defined as:

$$f(y_i) = \begin{cases} a_i y_i, y_i \leq 0 \\ y_i, 0 < y_i \leq 4 \\ b_i y_i, y_i > 4 \end{cases} \quad (6)$$

## EXPERIMENT AND ANALYSIS

### Dataset

The training and testing samples of painting images in this experiment come from https://www.robots.ox.ac.uk/~vgg/data/paintings/. It contains 8,629 pictures, including samples of similar paintings in categories such as flowers and birds, landscapes and figures, 3,463 training sets, 865 verification sets and 4,301 test sets.

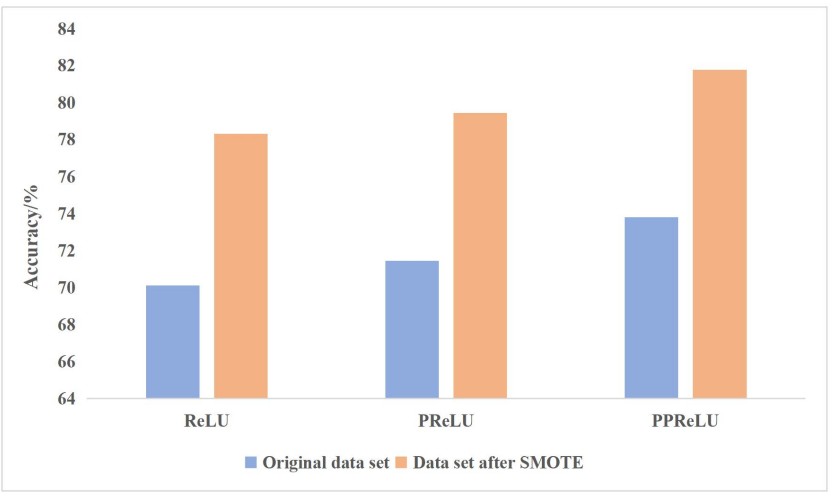

**Figure 4** Comparison of different shallow learning algorithms.

**Table 1** Comparison of training results of different algorithms (dataset 360 (60 × 6)).

|  | Accuracy/% | Convergence steps | Training time/s |
|---|---|---|---|
| CNN | 65.00 | 200 | 896.0 |
| NN | 78.00 | 220 | 30.8 |
| DBN | 61.60 | 200 | 30.0 |
| Ours | 66.33 | 170 | 943.5 |

## Experimental result

### Comparison of different activation functions

To verify the effectiveness of the improved activation function PPReLU, the CNN structure in Fig. 4 is adopted, and ReLU, PReLU and PPReLU are used as activation functions for training. The classification accuracy results are shown in Fig. 4.

In classifying painting images, the PReLU activation function is better than the ReLU activation function, and the newly proposed PPReLU activation function is better than the PReLU activation function. PReLU activation function supports negative numbers to pass through the activation layer, which solves the problem of partial gradient disappearance of negative numbers in ReLU.

### Comparison of different deep learning algorithms

Traditional CNN, NN, DBN and our model are used to test on the same data set. The experimental test data sets are the data before and after amplification, the training set before amplification is 2,340 (780×3), and the test set is 60 (20×3). The training set and test set after amplification are six times that before amplification, and the training set and test set are 14,040 (2340×6) and 360 (60×6), respectively. The experimental results are shown in Tables 1 and 2.

**Table 2  Comparison of training results of different algorithms (dataset 14,040 (2,340 × 6)).**

|       | Accuracy/% | Convergence steps | Training time/s |
|-------|------------|-------------------|-----------------|
| CNN   | 84.20      | 150               | 3,804.0         |
| NN    | 85.76      | 180               | 201.6           |
| DBN   | 73.94      | 180               | 201.6           |
| Ours  | 91.55      | 60                | 3,778.2         |

From Tables 1 and 2, in the proposed algorithm and the traditional CNN, NN and DBN, using the improved SMOTE amplified data can significantly improve the recognition rate of the learning method and the classification performance of the proposed method is superior to other methods, which proves the feasibility of the proposed method. However, after data expansion, the amount of data to be trained greatly increases, leading to a longer training time. Moreover, because data amplification can better extract comprehensive features, the convergence steps are reduced, and the classification performance is improved.

### Comparison of different shallow learning algorithms

In addition, the following shallow learning algorithms are used for comparison, SIFT+BOW (Scale-invariant Feature Transform+ Bag of Words) segmented the image into smaller and smaller sub-regions, calculated the local histogram features of each sub-region, then combined the features of all sub-regions, and used SVM for classification; HOG (histogram of gradient) is used to calculate the gradient histogram of local regions of images to form features, and SVM is used to classify features; compared with the deep learning method, LeNet adopts the multilayer artificial neural network method, and uses the backpropagation algorithm to update the weights for the first time; AlexNet adds the structural depth of convolutional neural network based on LeNet, and applies ReLU activation function and dropout to the network model. All experiments were conducted on the same data set. The comparison of experimental results between the traditional and deep learning methods is shown in Fig. 5.

The above comparison shows that the effect of the deep learning method in classifying painting images is better than that of traditional methods, especially for the enlarged data sets with relatively more classification categories and more incredible difficulty. LeNet and AlexNet adopt the traditional serial convolution neural network structure, so the classification effect is not as good as other methods based on feature extraction.

### Discussion

It can be seen from the above experimental results that the new activation function PPReLU, as an improved version of PReLU, suppresses the excessive value of the positive part through learnable parameters, reduces the variance of eigenvalues, makes the distribution of eigenvalues more concentrated, and thus accelerates the convergence speed of the network. Meanwhile, the gradient explosion caused by the extreme eigenvalues is prevented in the backpropagation process. The PPReLU activation function only introduces a few parameters, which generally do not bring training difficulties and over-fitting problems.

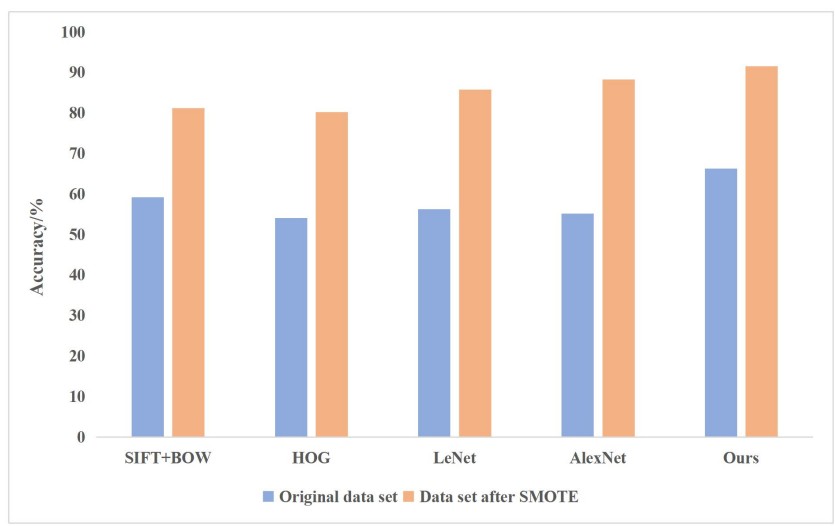

**Figure 5** Comparison of experimental results between the traditional and deep learning methods.

The introduced parameters also improve the network performance, which helps improve the accuracy of image classification.

In addition, data amplification through SMOTE can better extract comprehensive features, reduce the number of convergence steps, and improve classification performance. At the same time, the method in this article increases the width of the network structure based on the serial structure, realizes the extraction of various size features, and can more effectively extract the features of painting images and complete the image classification. Moreover, the method has a better classification effect on painting images compared with other methods.

This model realizes the effective extraction of different painting styles, which helps realize painting art therapy. On the one hand, painting in a non-verbal and symbolic way can help consultants understand the essence behind students' superficial problems more accurately and quickly and help consultants fundamentally solve students' problems and deepen the consulting effect; On the other hand, applying painting art therapy to mental health classroom can deepen the teaching goal of mental health education. Different classifications of painting styles can effectively help students know themselves from conscious to subconscious level in the shortest time in class and guide students to think about their roles, self-evaluation, interpersonal relationships and values. Furthermore, this study can help relieve anxiety in the process of painting and can also strengthen teamwork and improve interpersonal communication skills by using the form of group painting.

## CONCLUSION

Cyber-physical systems (CPS) are engineering architectures that combine virtual and actual processes. Adopting CPSs in imaging systems will significantly impact the development and interconnection of intelligent systems, such as online education, bioimaging, and

medical image processing. In modern education, mental health problems have become the focus and difficulty of students' education. As an effective mental health intervention, painting therapy has been integrated into the school's art education. This article uses a K-means algorithm to extract the colour distribution in a complete painting area. A convolution neural network is proposed to extract the features of painting images and realize the classification of painting images. Based on the traditional serial CNN network structure, a new convolution layer is connected in parallel between two convolution layers to increase the width of the network structure and realize multi-size feature extraction and multi-feature fusion. We find that the improved CNN model enhances feature learning ability by comparing different activation functions, deep learning methods and shallow learning methods. Thus, it can extract painting image features and complete image classification more effectively. At the application level, the model can effectively extract different painting styles, and its introduction into the mental health classroom can help to realize painting art therapy. It can enrich mental health education for college students, which plays a more active role in improving the actual effect of psychological education in colleges and universities.

### Funding
The authors received no funding for this work.

### Competing Interests
The authors declare there are no competing interests.

### Author Contributions
- Fake Ma conceived and designed the experiments, performed the computation work, authored or reviewed drafts of the article, and approved the final draft.
- Huwei Li performed the experiments, analyzed the data, prepared figures and/or tables, and approved the final draft.

### Data Availability
The dataset is available at https://www.robots.ox.ac.uk/~vgg/data/paintings/.
The code is available in the Supplementary File.

### Supplemental Information
Supplemental information for this article can be found online at http://dx.doi.org/10.7717/peerj-cs.1462#supplemental-information.

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
