# Peer review of "Online painting image clustering for the mental health of college art students based on improved CNN and SMOTE"

_PeerJ Computer Science, doi:10.7717/peerj-cs.1462_

## Round 0.1 · original submission · Major Revisions

Thanks for your submission, please carefully revise and resubmit the paper in light of the comments.

Reviewer 1 ·

Basic reporting

This paper first extracts the lost color of painting images based on K-means clustering, and then proposes a painting style classification model based on improved CNN. Then, the longitudinal depth and transverse width of the network are adjusted to optimize the structure of the CNN network. Compared with the traditional methods, the model proposed in this paper has higher accuracy in painting image style classification, which can provide a new idea for deepening the teaching objectives of mental health education. This paper is innovative and the writing logic is rigorous.

1. Firstly, the selection of keywords still needs to be improved. As one of the main innovations of this paper, the new activation function PPReLU needs to be emphasized in the keywords;

2. It is very important to apply painting art therapy to the mental health education of college students, especially in the context of the current COVID-19 epidemic, so I suggest adding the historical background of the COVID-19 epidemic in the introduction, which will help to better highlight the research significance of this paper;

Experimental design

3. In Section 3.1.1, the author gives the calculation formula of node priority, but the steps of color extraction seem not to be elaborated, so I hope the author can supplement this;

4. In addition, in Section 3.1.2, the author optimizes the selection of initial cluster centers through the "max-min criterion method". What is the "max-min criterion method"? The authors need to add a description to this, and add references if necessary;

5. I suggest the author to add a schematic diagram of CNN network structure in Section3.2.2. The current description is not detailed and intuitive;

Validity of the findings

6. In the analysis of Figure 5, Why does the author say "In order to effectively compare the classification effects of traditional methods and deep learning methods on eastern painting images and western painting images, "which does not seem to be relevant to the study in this paper;

7. In order to emphasize his own innovation, the author explains the development status of this research in detail, and I suggest the author to distinguish the achievements of others from his own contribution, so that the innovation points mentioned are vague and contradictory;

Additional comments

8. There are also many grammatical and spelling errors in the manuscript, which I would prefer to have corrected by a professional.

·

Basic reporting

Good

Experimental design

Good

Validity of the findings

Good

Additional comments

In this paper, based on the K-Means algorithm to extract the color distribution in the complete painting region, the convolutional neural network is proposed to extract the features of painting images, and the method of classification of painting images is realized. Based on the traditional tandem CNN network structure, a new convolutional layer is connected between two convolutional layers in parallel to increase the width of the network structure and realize multi-size feature extraction and multi-feature fusion. By comparing different activation functions, deep learning methods and melaleuca learning methods, we find that the improved CNN model enhances the feature learning ability of CNN, so as to more effectively extract features of painted images and complete image classification. This paper has some innovation, but it needs to be modified.

(1). The core content of this study is to use the improved CNN network to classify painting art styles, but the innovation of this study is still unclear, and the author needs to explain in detail the defects of traditional classification methods in the introduction.

(2). The author's writing idea in Section 2 seems to focus on two parts: painting image classification and CNN application, so I suggest to adopt secondary titles for this division;

(3). As for Figure 2, the author estimated the K value of k-means clustering by elbow method, but I suggest adding this part to the analysis of experimental results to enhance the logic of the article;

(4).Section 3.2.1 is titled "Improving SMOTE", but in my opinion, this method does not seem to solve the common overfitting problem of SMOTE;

(5). In the description of CNN network structure, it is necessary to make a schematic diagram of each connection layer, because the selection of CNN network structure will affect the performance of classification results;

(6). As presented, the writing is not acceptable for the journal. There are problems with sentence structure, verb tense, and clause construction;

(7). Cyber-physical systems seems not well reflected in the article, SMOTE and CNN did not reasonably deploy;

(8). The training and testing samples of painting images in this experiment come from the Gallery of Aesthetic Landscape Oil Paintings, readers would prefer a more detailed description;

(9). More discussion of the optimization of past methods and their embodiment in specific application scenarios.

---

## Round 0.2 · Minor Revisions

Dear Authors,

Thank you for submitting the updated version of the paper. However, only some improvements are needed to enhance the quality of your paper technically and to improve its wider readability. Specifically, the comparison table and updating the abstract. Therefore, Please revise and resubmit.

Reviewer 1 ·

Basic reporting

The revised version of the paper is improved much but there still few concerns to be addressed
1. As you have used the Deep learning to measure the mental health of students, Try to adjust that key work in the title of the paper.
2. Try to fix few language issues e.g. “However, traditional image classification models are prone to lose many detailed features”, what is lose in it?
3. Why Figure 4 and Figure 5 are same?

Experimental design

no comment

Validity of the findings

no comment

Additional comments

no comment

·

Basic reporting

Overall, paper's results and findings seem to be fine, but the abstract must be outlined in numerical form, e.g. the experimental results show that proposed model has the highest accuracy in classifying painting images style by comparing it with state-of-the-art methods can be quantitatively elaborated. This will increase abstract as well as paper's overall readability.

Experimental design

OK

Validity of the findings

OK

Additional comments

OK

---

## Round 0.3 · accepted · Accept

Thanks for addressing the comments of the experts, congratulations and good luck for your future research